# Association between Nuclear Morphometry Parameters and Gleason Grade in Patients with Prostatic Cancer

**DOI:** 10.3390/diagnostics12061356

**Published:** 2022-05-31

**Authors:** Kamil Malshy, Gilad E. Amiel, Dov Hershkovitz, Edmond Sabo, Azik Hoffman

**Affiliations:** 1Department of Urology, Rambam Health Care Campus, 8 HaAliya HaShniya Street, Haifa 3109601, Israel; k_malshy@rambam.health.gov.il (K.M.); g_amiel@rambam.health.gov.il (G.E.A.); 2Institute of Pathology, Sourasky Medical Center, Tel Aviv 6997801, Israel; dovh@tlvmc.gov.il; 3Department of Pathology, Carmel Medical Center, Haifa 3436212, Israel; edmondsa@clalit.org.il

**Keywords:** prostate cancer, cell morphometry, Gleason grade

## Abstract

Objective: Gleason scoring system remains the pathological method of choice for prostate cancer (Pca) grading. However, this method of tumor tissue architectural structure grading is still affected by subjective assessment and might succumb to several disadvantages, mainly inter-observer variability. These limitations might be diminished by determining characteristic cellular heterogeneity parameters which might improve Gleason scoring homogeneity. One of the quantitative tools of tumor assessment is the morphometric characterization of tumor cell nuclei. We aimed to test the relationship between various morphometric measures and the Gleason score assigned to different prostate cancer samples. Materials and Methods: We reviewed 60 prostate biopsy samples performed at a tertiary uro-oncology center. Each slide was assigned a Gleason grade according to the International Society of Urological Pathology contemporary grading system by a single experienced uro-pathologist. Samples were assigned into groups from grades 3 to 5. Next, the samples were digitally scanned (×400 magnification) and sampled on a computer using Image-Pro-Plus software^©^. Manual segmentation of approximately 100 selected tumor cells per sample was performed, and a computerized measurement of 54 predetermined morphometric properties of each cell nuclei was recorded. These characteristics were used to compare the pathological group grades assigned to each specimen. Results: Initially, of the 54 morphometric parameters evaluated, 38 were predictive of Gleason grade (*p* < 0.05). On multivariate analysis, 7 independent parameters were found to be discriminative of different Pca grades: minimum radius shape, intensity—minimal gray level, intensity—maximal gray level, character—gray level (green), character—gray level (blue), chromatin color, fractal dimension, and chromatin texture. A formula to predict the presence of Gleason grade 3 vs. grades 4 or 5 was developed (97.2% sensitivity, 100% specificity). Discussion: The suggested morphometry method based on seven selected parameters is highly sensitive and specific in predicting Gleason score ≥ 4. Since discriminating Gleason score 3 from ≥4 is essential for proper treatment selection, this method might be beneficial in addition to standard pathological tissue analysis in reducing variability among pathologists.

## 1. Introduction

Prostate cancer (Pca) is the second most common cancer worldwide, and the second cause of death from cancer in men [1,2]. Pca is highly subjected to over-treatment in case of localized, low grade (Gleason 6) disease. Localized Pca treatment options encompass a spectrum of treatments, including active surveillance, radical prostatectomy and radiation. Therefore, initial accurate tumor grading and staging are crucial for proper treatment selection.

Tissue diagnosis is commonly performed by a pathologist using microscopic examination of slides stained with haematoxylin and eosin. When Pca is diagnosed, the tumor grade is determined using the Gleason Scoring system (GS), based on tissue arrangement as determined by the pathologist, referring mainly to the prostate glandular arrangement and basic behavioral characteristics of cells within the stroma [3]. Gleason score is highly predictive of Pca treatment outcome and effective decision making since a higher Gleason score is considered to denote a potentially more aggressive Pca and suggests a worse prognosis [4].

The GS is exclusively used in the diagnosis and grading of prostate tissue tumors. However, it is accompanied by a significant degree of variation among pathologists who are responsible for determining an accurate diagnosis and the patient’s best treatment plan. Therefore, it is difficult to assign the same Gleason score due to the subjective nature of the GS [3,4,5]. To address these inter-and intra-observer discrepancies among pathologists, several updates to the GS were made [4]. Nevertheless, it was still reported that there was an up to 41% variability among different pathologists [5]. Additionally, the biopsy core may occasionally contain a small amount of tissue representative of the tumor, further expanding the risk of inter-observer variability. Attempts to reduce inter-observer variability have included computerized image analysis and machine learning [6]. Another suggested method to overcome the lack of objectivity in the conventional pathological analysis is a quantitative measurement of pre-determined pathological features of cancer cells [7]. Computerized nuclear morphometry (CNM) is considered a cost-effective, objective, and retrievable method for the evaluation of histological features [8]. This tool enables rapid measurement of parameters related to the size and shape of the nucleus; is considered an important prognostic indicator in breast cancer, renal cell carcinoma, and adenocarcinoma of the colon [9,10,11]; and is used in clinical practice [10]. The potential benefits of histological morphometric analysis are objectivity, accuracy, and efficiency. Computerized image analysis offers numerous morphometric characteristics and potentially can locate areas of malignant transformation in the tissue. However, because of the variance of prostate gland complexity and technical difficulties, analysis of the architectural complexity based on the calculation of fractal dimensions has not been incorporated into clinical practice.

In this study, we compared various morphometric parameters to pathological Gleason grades. We focused on intracellular basic parameters of prostate tumor behavior using objective methods, mainly in tissue biopsies containing a small amount of tumor.

## 2. Patients and Methods

### 2.1. Patients and Tissue Samples

We analyzed 60 achieved prostate biopsy samples harboring Pca of Gleason grades 1–3, 4, and 5. The Gleason grades were determined by a single uro-pathologist (A.S.) using the contemporary grading of Pca [4]. No patient was treated with androgen deprivation therapy before sample collection.

### 2.2. Data Acquisition

Each needle biopsy sample was analyzed in three steps:

Step 1: sample image capturing—Samples were scanned using a microscope (Olympus X 43, Tokyo, Japan) at a magnification of ×400. High-resolution tumor images were photographed with a digital camera (Retiga 2000 Qimaging, Barnaby, BC, Canada) and transferred to a computerized version 7.0 software mediated image-pro plus (Media Cybernetics, Rockville, MA, USA).

Step 2: segmentation—Morphometric characteristic measurements were produced for each sample manually (since tumor nuclei density was relatively high). While Image-Pro-Plus software enables automatic segmentation, separation capacity is limited and does not provide sufficient separation to allow proper analysis. Nuclear cell boundaries for each nucleus were marked, covering approximately 100 nuclei per sample. To reduce selection bias, this step was performed by a single person (K.M.), blinded to the samples’ Gleason scores. Next, the separation was completed by the ERL Image-Pro Plus program, based on the manually marked borders.

Step 3: morphometric characteristics measurements—With the aid of Image-Pro plus software, pre-determined characteristics were examined, including morphometric characteristics of tumor cell nuclei size (nucleus area, perimeter, diameter, and radius), shape (angle, axes, and roundness), and texture (chromatin density and heterogeneity).

### 2.3. Statistical Analysis

Morphometric analysis results are summarized as mean ± standard deviation. Morphometric variables of the tumor nuclear structure were compared between the three levels of Gleason using the one-way ANOVA test. In case of a significant difference of Gleason score grades, a Bonferroni test was used to correct for multiple variables. Next, variables showing significant differences were incorporated in a morphometric analysis test of gradual progress model (Wald stepwise forward method), ultimately selecting independent morphometric variables with a statistically significant association to Gleason score. We also used the discriminative analysis to calculate a regression coefficient in order to create a formula predictive of Gleason score. Additionally, we used the receiver-operating characteristic (ROC) curve to select the optimal point of sensitivity and specificity to predict the Gleason score for each case separately. *p* values ≤ 0.05 were considered to be statistically significant. All statistical analyses were processed using the SPSS 26.0^©^ (SPSS Inc., Chicago, IL, USA) software.

## 3. Results

### 3.1. General Characteristics of the Study Population

We examined 60 pathological samples of prostate adenocarcinoma, including 24 Gleason 1 to 3, 20 Gleason 4, and 16 Gleason 5. We used each tumor biopsy slide to obtain 4 to 10 microscopic images (depending on tumor area), of which we obtained a mean average of 5.4 ± 0.9 images for Gleason 1-3 slides, 4.4 ± 0.5, and 4.7 ± 0.8 for Gleason 4 and Gleason 5, respectively. Finally, 131.8 ± 28.5, 105.9 ± 15.2, and 104.7 ± 11.9 nuclei were detected, respectively.

### 3.2. Difference between Groups Using Univariant Analysis

Table 1 displays the univariant analysis of the various morphometric parameters analyzed.

Following 54 morphometric properties examined for univariant analysis, 38 were found to be statistically significant in predicting the Gleason score. Not all the variables were capable of distinguishing between all three levels (1–3, 4, and 5) of the Gleason score.

### 3.3. Indicators of Nuclear Size

We observed statistically significant separation capacity nucleus sizes (MNA—mean nuclear area) and circumference characteristics between the lowest Gleason score 1 to 3 and Gleason 5. However, these parameters were not distinguishable compared to Gleason 4. Still, the mean nuclear area demonstrated a gradual increase when comparing Gleason 1–3 to Gleason 4 and 5. Namely, 54.01 ± 7.48, 61.16 ± 17.069, and 71.14 ± 148 µm, respectively. An additional nuclear size characteristic, the minimal radius of the nucleus had a similar trend: 3.36 µm ± 0.29, 3.63 µm ± 0.5, and 3.72 µm ± 0.518 for Gleason 1–3, 4, and 5, respectively. This gradual trend of increase in size was also observed in other nucleus measurements (Table 1).

### 3.4. Variables Characterizing the Optical Density (Gray Levels)

Some optical density parameters were able to differentiate between Gleason 4 and Gleason 3 and 5, for instance, the “gray level of the green channel” was dark in the Gleason 1–3 nuclei (green-gray level = 97.079 ± 22.5), lighter in the Gleason 4 nuclei (114.22 ± 20.81) and again darker in Gleason 5 nuclei (89.46 ± 21.71). We speculated that this was related to the total amount of chromatin and therefore we calculated the integrated optical density (IOD = optical density × area) since the optical density is calculated as the opposite of gray level (optical density = 256–gray level). There was a gradual increase in IOD with higher Gleason grades, while a significant increase was observed between Gleason grades 4 and 5 (Table 1).

When applying a multivariate model of discriminative regression, seven independent variables of shape volume and nature of nuclei were shown to potentially differentiate between the three Gleason grades: minimal radius, minimum and maximum gray levels, green and blue gray level color of chromatin, the dimension of a fractal boundary object (fractal dimension), and chromatin texture (margination) (Table 2)**.** These variables illustrate the difference in size (radius), optical density (gray levels of the green and blue channels), texture (margination), and complexity of the nucleus wall (fractal dimension) of different Gleason score Pca cells (Table 3).

### 3.5. Discriminate Score

Based on the regression coefficients of the independent variables, a discriminate Score (DS) was calculated, as shown:**Discriminant Score (DS)** = 1331.50420234964 + (Minimal Radius × 5.95568628644) − (Fractal Dimension × 1479.27930311731) + (Gray Level (Min) × 0.40274897061) + (Gray Level (Max) × 0.53105523026) + (Margination × 275.38324653827) − (Gray Level (Green) × 0.89343166330) + (Gray Level (Blue) × 0.51289341737)

Next, based on the ROC, we found that an optimal DS score of −1.1134 was able to differentiate Gleason 1–3 and Gleason 4–5 with 97.2% sensitivity and 100% specificity (Figure 1).

## 4. Discussion

Pathological diagnosis of adenocarcinoma of the prostate is made by a pathologist using microscopic examination of tissue slides based on the glandular arrangement and basic properties of the cells in the tissue. Gleason Score remains the most reliable method to predict Pca prognosis [3,4,5,12,13,14]. However, reproducibility and agreement among pathologists remain a concern, mainly due to tissue artifacts and personal experience [5,7], subsequently affecting accurate Pca grading and choosing the proper treatment. In recent years, several methods have been reported to overcome the subjectivity of the conventional histological grading system by quantitative measurement of pathological features of cancer cells, potentially improving objectivity, accuracy, and efficiency [15,16,17]. Additional value is in samples containing a small volume of Pca within the tissue, which might not represent the true pathological grade. This challenge urged the attempts to seek morphometric characteristics (size, complexity, intensity, and texture painting chromatin) of the nuclei as predictors of pathological Gleason score.

### 4.1. The Relationship between the Increase in the Nucleus Size and Tumor Progression

Our method was able to produce a score capable to differentiate between pathological insignificant Pca (Gleason score 1–3) and higher grades that require treatment (Gleason 5) using the MNA and nuclear surface characteristics. However, these parameters do not differentiate between Gleason 1–3 and intermediate-risk Gleason 4, as well as between Gleason 4 and 5. Still, a trend of gradual increase in the mean nuclear area of each grade between Gleason scores 3 and 5 was observed. The minimum radius of the nucleus showed a similar trend (Table 1). This gradual increase was observed in additional measures of nucleus size, supporting findings reported by Bektas et al. in 2009, who examined 130 cases of Pca to study the relationship between Gleason score and nuclear morphometrics in 30 prostatectomy samples (only two with Gleason 5), and 100 prostate biopsies. A correlation between Gleason score and MNA in both prostatectomy and needle biopsies samples was reported [18].

### 4.2. The Relationship between Chromatin Density and Overall Quantity and Gleason Score

When examining optical density (gray level), we observed that chromatin density varies depending on the Gleason score. This optical density change is characterized as the “gray level of the green channel”, showing dark Gleason 1–3 nuclei, lighter Gleason 4 nuclei, and again darker Gleason 5 nuclei (Table 1). The gradual increase in nucleus size with higher Gleason score and simultaneously optical density brightening in Gleason 4 might be explained by a known phenomenon of tumors; namely, the increase in nucleus area and the appearance of vesicles (appearing as bright chromatin areas), which is related to the malignant behavior of the nuclei. As a more aggressive malignancy develops, darker nuclei are seen since the number of chromosomes and nuclear protein levels increase. Similarly, a correlation between optical density, aggressive malignancy, and lymph node metastasis has been reported [19]. While analyzing the relationship between the overall amount of cell chromatin and the Gleason score, a minimal difference in chromatin quantity between Gleason 3 and 4 was observed, but prostate tumor cells obtained from Gleason 5 areas contained a larger chromatin quantity, suggesting higher genomic instability (Table 1). As previously reported, while genomic instability is mainly related to aggressive prostate cancer, it could be used in early stages as well [20]. This relatively simple morphometric analysis may be further used to differentiate different Gleason score prostate cancer areas within the tissue.

### 4.3. Discriminant Score

Combining independent morphometric variables can predict the Pca tumor Gleason score with high precision (97.2% sensitivity and 100% specificity) (Figure 1). Objective intra-cellular parameters measurement may assist in determining Pca Gleason score particularly in tissue samples containing a small amount of Pca. However, morphometric analysis is currently time-consuming when compared to histological examination by an experienced pathologist. The use of the suggested discriminant score may not completely replace pathological analysis but may increase accuracy and reduce inter-observer discrepancies. Future use of specific cell membrane stains enabling the isolation of each cell from its neighbors and an automated morphometric tissue analysis might provide a powerful tool in assisting pathologists.

## 5. Conclusions

Size, complexity, nuclei staining intensity, and chromatin texture of prostate cancer cells can be used to distinguish between insignificant Pca (Gleason score 1 to 3) and significant Pca (Gleason 4 and 5). Improving accuracy and reducing inter-observer variability of Gleason scoring can be achieved by computerized mathematics of selected intra-cellular characteristics, allowing better standardization of the Gleason grading system. Additional validation in a larger cohort of patients and tissue samples is required.

## Figures and Tables

**Figure 1 diagnostics-12-01356-f001:**
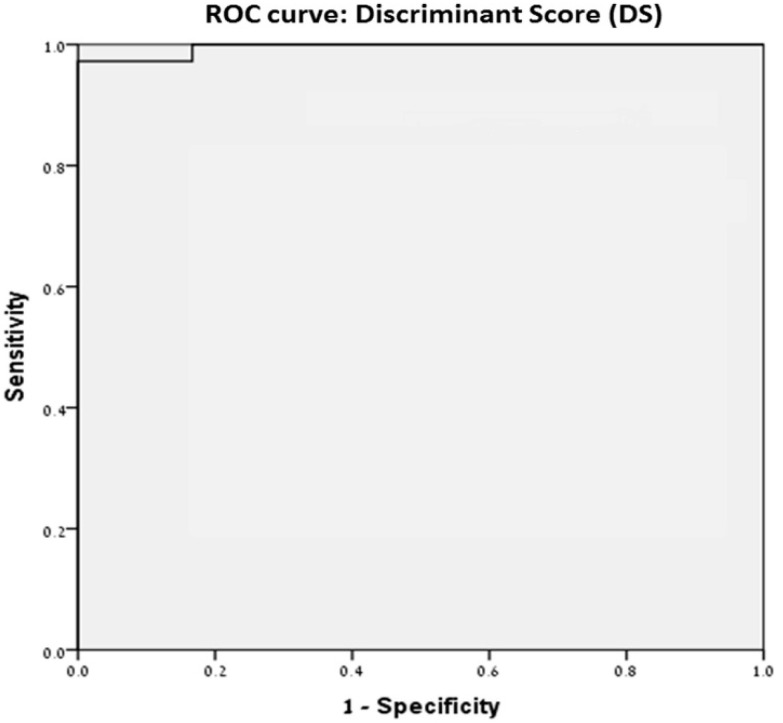
**Receiver-operating curve of selected morphometric characteristic.** A discriminate score of −1.1134 can distinguish between low Gleason grade 3 and high Gleason score (4 and 5) with 97.2% sensitivity and 100% specificity.

**Table 1 diagnostics-12-01356-t001:** Morphometric comparisons between the different levels of Gleason (univariate analysis).

Morphometric Variant	Description	Illustration	GleasonGrade 1–3	GleasonGrade 4	Gleason Grade 5	*p*-Value (ANOVA)	Bonferroni Test
Nuclear Area	The area contained in the polygon encloses the object	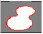	54.01 ± 7.48	61.16 ± 17.069	71.14 ± 148	0.006	3 vs. 5
Nuclear Aspect	The relation between axial and ellipse axes to the object	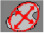	1.28 ± 0.072	1.22 ± 0.048	1.32 ± 0.10	0.000	3 vs. 44 vs. 5
Area/Box	The ratio of the area to the area of an enclosing box	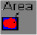	0.745 ± 0.011	0.75 ± 0.006	0.741 ± 0.013	0.022	4vs.5
Gray Level-Mean	The average density or intensity of an object	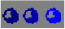	126.37 ± 17.64	150.84 ± 17.87	122.10 ± 22.09	0.000	3 vs.4/4 vs.5
Axis Major	The length of the main axis of an ellipse	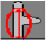	9.28 ± 0.61	9.60 ± 1.32	10.75 ± 1.77	0.002	3 vs.5/4 vs.5
Axis Minor	The length of thesecondary axis of anellipse equals torque	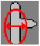	7.34 ± 0.59	7.91 ± 1.078	8.199 ± 1.126	0.016	3vs.5
Maximal Diameter	The length of the longest line passing through the center	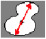	9.26 ± 0.618	9.60 ± 1.34	10.733 ± 1.79	0.002	3 vs.5/4 vs.5
Minimal Diameter	The length of theshortest line passing through the center	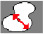	7.02 ± 0.589	7.58 ± 1.043	7.83 ± 1.079	0.019	3vs.5
Mean Diameter	Average of locomotives passing through the center	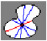	8.06 ± 0.568	8.53 ± 1.16	9.14 ± 1.34	0.008	3vs.5
Maximal Radius	Maximum distance from center to outline	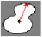	4.809 ± 0.319	4.97 ± 0.69	5.59 ± 0.945	0.002	3 vs.5/4 vs.5
Minimal Radius	Minimum distance from center to outline	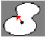	3.36 ± 0.29	3.63 ± 0.5	3.72 ± 0.518	0.027	3vs.5
Holes	Number of holes in the object	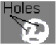	26.36 ± 1.811	27.72 ± 3.83	30.199 ± 4.66	0.005	3vs.5
Radius Ratio	The ratio between the maximum radius and the minimum	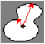	1.466 ± 0.11	1.37 ± 0.0578	1.529 ± 0.139	0.000	3 vs.4/4 vs.5
Nuclear Roundness	Roundness according to the ratio between the outline and the surface	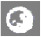	1.088 ± 0.027	1.06 ± 0.011	1.096 ± 0.027	0.001	3 vs.4/4 vs.5
Gray Level–Red	Average red value in object	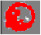	146.34 ± 21.48	183.02 ± 29.21	140.06 ± 36.48	0.000	3 vs. 4/4 vs. 5
Gray Level–Green	Average green value in object	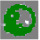	97.079 ± 22.5	114.22 ± 20.81	89.46 ± 21.71	0.03	3 vs. 4/4 vs. 5
Gray Level–Blue	Average blue value in object	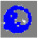	135.68 ± 14.29	155.28 ± 13.96	136.79 ± 15.39	0.000	3 vs. 4/4 vs. 5
Length	Evaluation of object length by descriptor	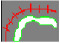	9.28 ± 0.616	9.60 ± 1.33	10.76 ± 1.80	0.002	3 vs.5/4 vs.5
Width	The width of the object is weighted by thecurvature of a structure	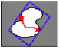	7.38 ± 0.587	7.94 ± 1.09	8.26 ± 1.159	0.015	3vs.5
Perimeter2	The chain code for the outline includes holes	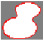	27.859 ± 1.925	29.357 ± 4.119	31.97 ± 4.889	0.004	3vs.5
IOD	The optical density of the field	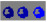	6937.39 ± 1578	9426.03 ± 3208	8796.673 ± 3221	0.009	3vs.4
Perimeter (convex)	The length of theconvex outline of the object	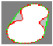	26.074 ± 1.80	27.42 ± 3.81	29.876	0.005	3 vs. 5
Perimeter (ellipse)	Equivalent ellipseoutline length	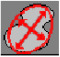	26.23 ± 1.81	27.60 ± 3.75	29.95 ± 4.531	0.005	3 vs. 5
Perimeter (ratio)	The ratio of the convex outline to the outline of the object	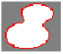	51.69 ± 7.31	58.718 ± 16.75	68.49 ± 22.10	0.006	3 vs. 5
Fractal Dimension	Fractal dimension of the object boundary	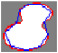	1.058 ± 0.00159	1.057 ± 0.0017	1.056 ± 0.0013	0.002	3 vs. 5
Box Width	Width of enclosing box	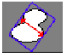	8.54 ± 0.656	9.01 ± 1.141	9.7229 ± 1.50	0.006	3 vs. 5
Box Height	Height of enclosing box	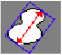	8.40 ± 0.6106	8.82 ± 1.34	9.63 ± 1.568	0.008	3 vs. 5
Feret (min)	The shortest distance in the defined direction	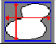	7.277 ± 0.58	7.82 ± 1.077	8.14 ± 1.135	0.016	3 vs. 5
Feret (mean)	Average distances in different directions	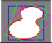	8.383 ± 0.576	8.816 ± 1.21	9.596 ± 1.47	0.005	3 vs. 5
Gray Level (min)	Density or minimal power in the object	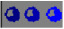	85.54 ± 13.98	102.94 ± 15.25	82.446 ± 20.027	0.000	3 vs. 4/4 vs. 5
Gray Level (max)	Density or maximum intensity in the object	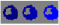	178.70 ± 15.86	205.66 ± 15.05	180.61 ± 18.85	0.000	3 vs. 4/4 vs. 5
Gray Level (std.dv)	Standard Deviation The density or intensity of an object	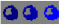	17.53 ± 2.4255	20.12 ± 3.188	17.88 ± 2.03	0.005	3 vs. 4/4 vs. 5
Margination	Dissemination of the intensity ratiosbetween center andperimeter	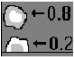	0.3400 ± 0.015	0.334 ± 016	0.352 ± 0.0158	0.004	4 vs. 5
Heterogeneity	Percentage of pixels that deviate above 10%	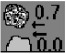	0.148 ± 0.0154	0.208 ± 0.0176	0.152 ± 0.0447	0.004	3 vs. 4/4 vs. 5
Clumpiness	Reflects texturevariation	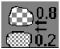	0.122 ± 0.0163	0.178 ± 0.107	0.110 ± 0.061	0.025	4 vs. 5
Gray Level (sum)	Total density or optical intensity	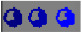	270125 ± 61457	367027 ± 12494	342521 ± 12542	0.009	3 vs. 4
Perimeter3	Corrected chain code for outline, without holes	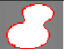	26.270 ± 1.811	27.611 ± 3.82	30.09 ± 4.66	0.005	3 vs. 5
Perimeter Length	Evaluation of object length by layout	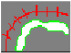	13.135 ± 0.90	13.80 ± 1.91	15.14 ± 2.333	0.005	3 vs. 5

**Table 2 diagnostics-12-01356-t002:** Multivariate analysis results.

Independent Variable	Beta, Slope (Incline)	*p* Value
Minimal Radius	5.9556	<0.0001
Gray Level (Green)	−0.8934	<0.0001
Gray Level (Blue)	0.512	<0.0001
Fractal Dimension	−1479.2	<0.0001
Gray Level (Min)	0.4027	<0.0001
Gray Level (Max)	0.531	<0.0001
Margination	275.38	<0.0001
Point of intersection with the Y-axis (Constant)	1331.5	

**Table 3 diagnostics-12-01356-t003:** Total amount of chromatin quantity depending on the Gleason score.

Gleason Grade	Gray Level Green	Optic Density	Nuclear Area	Total Chromatin Quantity
**3**	97	256 − 97 = 159	54	8586
**4**	114	256 − 114 = 142	61	8662 *
**5**	89	256 − 89 = 167	71	11,857 *

** p* < 0.05.

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
