# Peer review of "Association between Nuclear Morphometry Parameters and Gleason Grade in Patients with Prostatic Cancer"

_diagnostics, 2022, doi:10.3390/diagnostics12061356_

Round 1

Reviewer 1 Report

The authors of the study investigate the association between nuclear morphometry parameters and Gleason grade in patients with prostatic cancer. The study reviews 60 radical prostatectomy specimens with assigned Gleason grade by a genitourinary pathologist. Samples are digitally scanned and manual segmentation of approximately 100 tumor cells per sample was performed with 54 predetermined morphometric properties for each cell nuclei.  These characteristics were used to compare the Grade Group is assigned to each specimen.  7 independent parameters were found to be discriminatory between different prostate cancer grade which included: Minimum radius shape, intensity of minimal gray level, intensity of maximal grade level, character of grade level (green), character of grade level 2 (blue), chromatin color, refractile dimension and chromatin texture.  A formula was generated to predict the presence of Gleason grade 3 versus grades 4 or 5 which showed 97.2 sensitivity and 100% specificity.  Based on these results, the authors suggest morphometry method on the selected parameters is highly sensitive and specific in predicting Gleason score greater than or equal to 4.  The authors emphasized that discriminating Gleason score 3 from greater than Gleason score 4 is essential for proper treatment selection.

Overall, the study design and extensive analyses are well thought out and presented.  I believe such findings in the context of the development of integration with AI and pathology may be extremely useful.  Several studies have shown that not all Gleason pattern 4 designations behave equally or associated with poor outcomes.  It would be interesting to see if the authors noted any distinct morphologic pattern differences in the Gleason score 4 designation group.  In particular, the distinction between a poorly formed gland pattern 4 versus glomerulation versus cribriform morphologies.

In summary, this is an interesting study that provides extensive data analyses and data points with the generation of a formula to predict the presence of Gleason grade 3 versus grades 4 and 5 with high sensitivity and specificity.

Reviewer 2 Report

In this retrospective study the authors are attempting to if nuclear morphometry parameters can correlate with gleason scoring system.

The findings are interesting and the analyses are well performed.

Comments:

In the abstract, the authors mentioned that the prostate cancer samples were obtained from radical prostatectomy specimens, however in the manuscript they mentioned that the prostate cancer samples were from prostate biopsies (line 82). The authors should be more clear regarding this point.

The authors should mention in the methods if any patient received preoperative treatment like ADT or radiation. If not, the authors should also mention that none had received pre op hormonal therapy or radiation.

In the introduction, I suggest removing this sentence “ while localized prostate 39 cancer has an excellent prognosis, metastatic Pca disease has a 5-year survival rate of 28%.”

Because not all localized pca have excellent prognosis! and the 5-y survival of 28% in mPCa has no refernce, and might no be accurate because of the heterogenous nature of pca.

In the introduction, lines 42-44

this statement is wrong, please just mention the types of treatments without the indication (for example radical prostatectomy and radiation are not only for aggressive tumors and lesion targeted therapies are not only for low-risk diseases (especially that no definition is provided for low-risk disease)

Line 64: error, “this lack of subjectivity “should be “this lack of objectivity”

First sentence of discussion: add a reference

Line 186: “affecting accurate Pca staging” should be grading not staging

Reviewer 3 Report

This study was reported the association between nuclear morphometry parameters and Gleason grade in patients with PCa. Overall, this manuscript is very interesting and well written. The reviewer would like to suggest some critiques as follows.

Major revision

  1. The reviewer thinks that Gleason grade 1 is an indolent cancer. However, prostate cancer with Gleason grade ≥2 may have several disadvantages because of occasionally aggressive behaviors such as invasion or distant metastasis. Therefore, the reviewer thinks that the authors may investigated the association between nuclear morphometry parameters and Gleason grade in patients with PCa who had Gleason grade ≥2.

Author Response

Point 1

The reviewer thinks that Gleason grade 1 is an indolent cancer. However, prostate cancer with Gleason grade ≥2 may have several disadvantages because of occasionally aggressive behaviors such as invasion or distant metastasis. Therefore, the reviewer thinks that the authors may investigated the association between nuclear morphometry parameters and Gleason grade in patients with PCa who had Gleason grade ≥2.

Response 1:

We agree that Gleason grade 1 requires only active surveillance. However, the main concern in Group grade 1 disease, especially in high volume Gleason 6 samples, is misdiagnosis, partially due to incorrect pathological grading. Our suggested method aims to address this issue.

Our future goal is to better differentiate adverse morphologies of Gleason 4 (GG2 and GG3) disease since it has a major impact on treatment selection.